# A Modified 3D-QSAR Model Based on Ideal Point Method and Its Application in the Molecular Modification of Plasticizers with Flame Retardancy and Eco-Friendliness

**DOI:** 10.3390/polym12091942

**Published:** 2020-08-28

**Authors:** Haigang Zhang, Chengji Zhao, Hui Na

**Affiliations:** Alan G. MacDiarmid Institute, College of Chemistry, Jilin University, No. 2699 Qianjin Street, Changchun 130000, China

**Keywords:** plastics, flame retardancy, phthalic acid esters (PAEs), eco-friendliness, three-dimensional quantitative structure-activity relationship (3D-QSAR), molecular modification

## Abstract

The addition of plasticizers makes plastics flammable, and thus, poses a potential risk to the environment. In previous researches, plasticizers with flame retardancy had been synthesized, but their eco-friendliness had not been tested or described. Thus, in this paper, eco-friendliness plasticizers with flame retardancy were designed based on phthalic acid esters (PAEs), which are known as common plasticizers and major plastic additives. For a comprehensive analysis, such as flammability, biotoxicity, and enrichment effects, 17 PAEs’ comprehensive evaluation values were calculated based on the ideal point method. Further, a multi-effect three-dimensional quantitative structure-activity relationship (3D-QSAR) model of PAEs’ flammability, biotoxicity and enrichment effects was constructed. Thus, 18 dimethyl phthalate (DMP) derivatives and 20 diallyl phthalate (DAP) derivatives were designed based on three-dimensional contour maps. Through evaluation of eco-friendliness and flammability, six eco-friendly PAE derivatives with flame retardancy were screened out. Based on contour maps analysis, it was confirmed that the introduction of large groups and hydrophobic groups was beneficial to the simultaneous improvement of PAEs’ comprehensive effects, and multiple effects. In addition, the group properties were correlated significantly with improved degrees of the comprehensive effects of corresponding PAE derivatives, confirming the feasibility of the comprehensive evaluation method and modified scheme.

## 1. Introduction

Numerous properties of plastic materials, lightweight features, corrosion resistance ability, bright color, transparency, ease to process, etc., have made them greatly applicable in many fields. Furthermore, the versatility, low cost, and abundance of plastic-based products also influence their daily applicability [1]. Due to being constructed by organic molecules, plastics are rich with ignition properties, which contribute to a potential fire-risk [2]. Due to the highly flammable property of plastic materials, they have a restricted use in high-temperature areas. Therefore, to avoid the significant loss of lives and properties caused by fire disaster, it is necessary to reduce the flammability of plastics through appropriate flame retardant treatment (such as covering the surface with a flame retardant coating and adding flame retardants) [1,3]. However, the addition of flame retardants is one of the prime methods to eliminate the flammability of plastics. So far, many flame retardants have been used to improve the flame retardant performance of plastics, which includes inorganic, halogenated, and organo-phosphorus flame retardants, as well as intumescent flame retardants. Moreover, the plastic compositions contain a high proportion of flame retardants, i.e., 10–20% [2,4].

Plastic polymers are durable and exhibit strong resistance to degrade in nature; besides, due to high consumption and low recycling ratio, plastic content is increasing limitlessly in the environment. In 2016, 335 million tons of plastic were produced worldwide [5], while in 2017, the amount of production increased up to 348 million tons [6], which indicates, the rate of plastic production has grown by about 9% per year, globally. In general, plastics are not easily biodegradable, but it is possible to decompose them gradually through mechanical action. Natural processes, such as ultraviolet-B (UV-B) radiation, atmospheric oxidation, pyrolysis, and seawater hydrolysis, fragment plastic into small pieces and convert them to microplastics (MP) (0.1–5000 µm), or even nanoplastics (NP) (<0.1 µm) [5,7]. MPs and NPs pollute and destroy the environment by releasing additives such as plasticizers [8]. The plastic wastes that get discharged into the environment, not only harmful to human health but also affect terrestrial and aquatic life, as well as the ecological system [9]. Moreover, MPs can interact with biological organisms; they are capable of penetrating through the tissues, accumulate in various organs, expose throughout the organisms, and thereby cause potent toxic effects on the whole living system [10]. MPs have caused serious impacts on marine microorganisms, that include decreasing feed rate, decrease in predation ability, physical damage, induced oxidative stress response, affected reproduction, neurological dysfunction, oxidative damage, and increased mortality [5]. Ingestion of MPs can cause adverse effects (such as mechanical damage, blockage of the digestive tract, affecting growth and development) on fish, and also involve some potentially harmful substances (such as plastic additives) into the aquatic food chain [11]. However, in-depth research on the adverse effects of MP on freshwater organisms, terrestrial organisms, and human bodies is still lacking in comparison to other fields. Lu et al. [12] reported that MPs can induce dysregulation of intestinal microflora in mice and lead to disorders of liver lipid metabolism. The human body may be exposed to plastics through diet, by inhalation of NP-containing aerosols, and also via skin contact [9]. In a similar way, MPs may also be enriched in the body up to a certain degree. As per the MP-enriched aquaculture studies performed by Wu et al. [13], all three typical aquatic organisms (fish, bivalves, shrimps) accumulate MPs, while the potential of microplastic accumulation of shrimp was lower than the other two species.

Phthalic acid esters (PAEs), commonly known as phthalates, are widely used as plasticizers in plastic industries [14], with up to 90% of the plasticizers used in China alone [15]. Although plasticizers improve the strength and flexibility of plastic products significantly [16], the addition of several plasticizers increases the flammability of plastic polymers, which eventually restricts applicability [17]. Therefore, eco-friendly, biodegradable, and flame retardant plasticizers are highly demanding in the industries [16]. The limited oxygen index (LOI) is the most common indicator to measure the flammability of a substance. LOI may be defined as the lowest level of oxygen concentration (expressed as a volume percentage) that support a substance burns in a uniformly flowing mixture of oxygen and nitrogen; under such condition, the part below the flame will not burn or meltdown further (candle-light) [17,18].

PAEs exhibit several harmful effects on the growth and development of human, animals (such as fish, shrimp), plants (such as algae), and planktons, such as embryotoxicity, neurotoxicity, genotoxicity, etc. [19] Due to estrogen toxicity by PAEs, several adverse effects may happen, which include multiple hormone disorders, affected endocrine, skeletal changes, male sperm reduction, and reproductive dysfunctions. Long-term exposure to PAEs is associated with a risk of cancer at the reproductive system [20]. Poopal et al. [21] monitored abnormal behavior and circadian rhythm disturbances in zebrafish that is caused by diheptyl phthalate (DHpP) and diisodecyl phthalate (DIDP). Similar to plastics (MPs and NPs), some PAEs are characterized by bio-enrichment properties that can accumulate in organisms through the food chain [22]. Jarosova et al. [23] studied the enrichment of dibutyl phthalate (DBP) and diethylhexyl phthalate (DEHP) in different tissues of chicks, and found the accumulation of DEHP in muscle, adipose tissue and skin were 3.2, 2.6 and 2.9 times higher than DBP.

In summary, plastics and its main additive PAEs are consistent in exhibiting several adverse effects in the environment and eco-system that include flammability, biotoxicity, and enrichment. On the other hand, the application of many flame retardant additives and their byproducts cause serious health hazards and environmental issues, which limit their industrial application [14,24]. Due to the same reason, development of low toxic and eco-friendly flame retardant technologies has become a very important research direction. In this paper, PAEs were used as the prime research objects to obtain information about the flammability property, biotoxicity, and enrichment effects of plastics, according to which new eco-friendly PAE-based plasticizers with flame retardancy were developed. In order to evaluate the molecules comprehensively, it is necessary to treat a variety of indicators with different dimensions using mathematical methods or equations. For example, Vahabi et al. [25,26] defined a simple universal dimensionless index for the first time, known as the flame retardancy index (FRI), which was useful for the comparative evaluation of the flame retardancy performance of thermoplastic systems regardless of the types of polymers and additives used. Li et al. [27] evaluated molecules’ multi-effect comprehensively via a simplified formula method and standard deviation score method. In this paper, dimensionless processed activity data of the flammability, biological (fish) toxicity and enrichment effects of 17 PAEs were evaluated via mathematical methods (ideal point method), which further combined to calculate the multi-effect comprehensive evaluation values of 17 PAEs. Then, a three-dimensional quantitative structure-activity relationship (3D-QSAR) model of PAEs’ flammability, biotoxicity, and enrichment multi-effect was constructed by taking the PAEs’ flammability, biotoxicity and enrichment multi-effect comprehensive evaluation values as the dependent variables and the molecular structure parameters as the independent variables. To obtain PAE derivatives with low biotoxicity, low enrichment, and flammability, the target molecules dimethyl phthalate (DMP) and diallyl phthalate (DAP) were modified by single and double substituent groups through the three-dimensional contour map of the multi-effect model. In addition, through the prediction and evaluation of PAEs’ multi-effect 3D-QSAR model, flammability, biotoxicity, enrichment, degradability, and mobility single-effect 3D-QSASR models, it was confirmed that the designed PAE derivatives were eco-friendly molecules with the better flame retardant property. At the same time, the stability and insulation of PAE derivatives were also evaluated.

## 2. Materials and Methods

### 2.1. Sources of Data

Most PAEs are used as plasticizers in the processing of polyvinyl chloride (PVC) [27]. Referring to Li et al. [27], 17 kinds of PAEs from PVC plastics were selected as representatives, which consist of higher molecular weight molecules and lower molecular weight molecules, long side-chain molecules and short side-chain molecules, different structural molecules including ring, linear chain or branched chain. The flammability property of 17 kinds of PAEs was characterized by LOI. Greater LOI value indicates the requirement of a higher amount of ultimate oxygen concentration for the combustion of material, and subsequently confirms the better flame retardant capability [18]. The flame retardant property of polymer materials is related to the molecular structure of the polymer itself. Empirical Formulas (1) and (2), which are applicable for halogen-free materials to calculate the ratio between the residue amount and flame retardancy during thermal decomposition, are commonly used to calculate the LOI value [28]. Following the same rule, here in this research work, the LOI data of PAEs were obtained from the calculations of Formulas (1) and (2) (Table 1). The biotoxicity data of 17 PAEs was obtained through the Estimation Program Interface (EPIWEB) 4.1 (Copyright© 2020–2012, United States Environmental Protection Agency for EPI Suite^TM^ and all component programs except BioHCWIN and KOAWIN, U.S.) software and employed as the 50% lethal concentration of PAEs to fish (lethal concentration of 50%, LC_50_) (Table 1). Bio-concentration factors (log*BCF*) of 17 PAEs were predicted by EPIWEB software to characterize the molecular enrichment (Table 1). Half-life (HL) data of 17 PAEs in rivers were introduced from the EPIWEB software; the logarithm value of the same (log*HL*) was used as an indicator of degradation abilities of the PAEs (Table 1). EPIWEB software was used to obtain PAEs’ experimental or predictive values for octanol/air partition coefficient (log*K*_OA_) to represent their migration (Table 1). In addition, the hydrophobicity parameter octanol/water partition coefficient (log*P*) of PAEs derivatives was calculated by software Chembiodraw 12.0.
(1)LOI=17.5+0.4CR
where CR refers to the amount of residue when the substance is heated at 850 °C.
(2)$$\mathrm{CR}=\frac{1200\left(\sum {\left(\mathrm{CFT}\right)}_{i}\right)}{M}$$

In the formula, (CFT)_i_ is the contribution index of the ith functional group to the residue amount of a substance. The CFT value of the functional group 
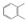
 contained in PAEs is 2, while the CFT values of other halogen-free functional groups are 0. M refers to the molecular weight of monomers in the polymer, and here refers to the relative molecular weight of PAEs.

### 2.2. Multi-Effect Comprehensive Evaluation Method for PAEs’ Flammability, Biotoxicity, and Enrichment—Ideal Point Method

To eliminate the influence of the index dimension on molecular evaluation, first, the performance indicators of PAEs’ flammability, biotoxicity, and enrichment should be processed dimensionless using Formulas (3) and (4) via normalization method, which is the superior way for the comprehensive evaluation of PAEs. The normalized index data C_ij_, is processed by the ideal point method (Formula (5)) [29]; further, the comprehensive evaluation value Z of PAEs’ flammability, biotoxicity and enrichment was calculated by weighting them together, with the weight ratio of PAEs’ flammability, biotoxicity, and enrichment at 40%:30%:30%. The above step should be performed before constructing a 3D-QSAR model of PAEs’ flammability, biotoxicity, and enrichment multi-effect, so as to evaluate the PAEs comprehensively.
(3)$$C_{\mathrm{ij}}=\frac{X_{\mathrm{ij}}-{\mathrm{minX}}_{j}}{{\mathrm{maxX}}_{j}-{\mathrm{minX}}_{j}}$$
(4)$$C_{\mathrm{ij}}=\frac{{\mathrm{maxX}}_{j}-X_{\mathrm{ij}}}{{\mathrm{maxX}}_{j}-{\mathrm{minX}}_{j}}$$
where C_ij_ refers to the ith dimensionless parameter of the jth index; maxX_j_ and minX_j_ are the maximum and minimum values of the jth index, respectively; for the positive indexes, it is applicable to Formula (3); for the inverse indexes, Formula (4) is applicable.
(5)$$Z_{i}=\overset{3}{\underset{j=1}{\sum }}\lambda_{j}|C_{\mathrm{ij}}-C_{j}^{*}|$$
where λ_j_ is the weight of jth index, among which λ_1_ = 40%, λ_2_ = 30%, λ_3_ = 30%; C_j_^*^ is the ideal point value of the jth index. After normalization of PAEs’ effect index parameters, 0 ≤ C_ij_ ≤ 1, and all of them were positive indexes; therefore, the value of C_j_^*^ in the equation is 1. Z_i_ is called Euclidian distance, namely the comprehensive evaluation value of the ith PAEs’ flammability, biotoxicity, and enrichment.

### 2.3. Construction of PAEs’ Flammability, Biotoxicity, and Enrichment Multi-Effect 3D-QSAR Model

The three-dimensional structures of PAEs were drawn in Sybyl-X 2.0; molecular geometry was mechanically optimized to obtain the low-energy conformation of each molecule using the Powell method followed by the Conjugate Gradient method while the Minimize module was employed. The molecular charge was set as Gasteiger-Huckle. Molecular force field Tripos was exerted on molecules, and the energy convergence standard was set as 0.005 kcal·mol^−1^ with 10,000 iterations.

The most active molecule was selected as the template molecule, while the Align Database module was used to superposition the skeleton of the compounds. During the calculation of the parameters of the Comparative Molecular Similarity Index Analysis (CoMSIA) field, the types of molecular force fields were selected as steric field (S), hydrophobic field (H), electrostatic field (E), hydrogen bond donor field (D), and hydrogen bond acceptor field (A), to gain the molecular field parameters. The dielectric constant was related to the distance when the threshold value was set as 125.4 kJ/mol. The remaining parameters adopted the default values of the system. CoMSIA field parameters could be calculated automatically via Sybyl-X 2.0 by entering PAEs’ Eigenvalues of the training set. When the Partial Least Square (PLS) method was used to analyze the relationship between the molecular structure and biological activity [30], the Leave-One-Out (LOO) method was used to cross-validate the training set compounds initially [31], which is required to estimate the cross-validation coefficient q^2^ and the optimal number of principal components *n*. Thereafter, “No Validation” was used for regression analysis while r^2^, standard error of estimate (SEE), and Fischer’s value (F) were calculated to complete the construction of the CoMSIA model [32]. In addition, the CoMSIA model needs to be verified externally by external test sets. The interactive validation coefficient r^2^_pred_ and standard deviation standard error of predict (SEP) of the test sets were used for confirming the external prediction ability of the model [33,34,35].

### 2.4. Evaluation of Eco-Friendliness, Stability, and Insulation of PAE Derivatives

In this paper, the single-effect 3D-QSAR models of PAEs’ biotoxicity, enrichment, degradability, and mobility were established, respectively, for testing the eco-friendliness of PAE derivatives. The single-effect CoMSIA models of PAEs’ biotoxicity, enrichment, degradability, and mobility were constructed by considering PAEs’ structure parameters as independent variables and PAEs’ log*LC*_50_, log*BCF*, log*HL* and log*K*_OA_ values as dependent variables. To screen the eco-friendliness of PAE derivatives, log*LC*_50_, log*BCF*, log*HL* and log*K*_OA_ values of the designed PAE derivatives were predicted using the above model.

Using software Gaussian 09, the structure of PAEs and its derivatives was optimized at the level of B3LYP/6–31G* base group via density functional theory (DFT). The positive frequency value (representing stability) and energy gap value (representing insulation) were calculated subsequently, while the results of the same were viewed by GaussView 5.0 and Multiwfn. When the positive frequency (Freq.) of PAE derivative exhibits a value greater than zero, it indicates the molecular structure of the designed PAE derivative is stable and can exist in a stable form in the environment [36]. The energy difference between the highest occupied molecular orbital (HOMO) and the lowest unoccupied molecular orbital (LUMO) is called Energy gap, which refers to the transition ability of the electrons from an occupied orbital to vacant orbital; this is an important parameter reflecting the electrical conductivity and luminescence properties of materials. A greater energy gap value indicates a weaker conductivity of the molecule [37].

## 3. Results

### 3.1. Construction and Verification of Multi-Effect 3D-QSAR Model of PAEs’ Flammability, Biotoxicity, and Enrichment

In this paper, a 3D-QSAR model of PAEs’ flammability, biotoxicity, and enrichment multi-effect was constructed by considering the molecular structure parameters of PAEs as independent variables and the comprehensive evaluation value Z of PAEs’ flammability, biotoxicity and enrichment multi-effect as dependent variables.

The training set is used to construct the PAEs’ multi-effect 3D-QSAR model, and the test set is used to verify its accuracy. According to the ratio of 3:1, 13 PAEs were randomly selected as the training sets, and the remaining four PAEs were used as the test sets. DMEP was used as the template molecule to build the multi-effect 3D-QSAR model of PAEs’ flammability, biotoxicity, and enrichment; the results are shown in Table 2. The evaluation parameters and molecular force field contribution rate of PAEs’ multi-effect CoMSIA model are shown in Table 3.

As per the results in Table 3, the optimal principal component *n* of PAEs’ multi-effect CoMSIA model was 10, while the cross-validation coefficient q^2^ was 0.616 (generally considered that when the value of q^2^ > 0.5, the model has the credible predictive ability [38,39]). The non-cross-validation coefficient r^2^ was 1.000 (>0.9), SEE was 0.008, and F was 507.314, indicating the greater fitting ability and robustness of the model [40,41]. The external test coefficient r^2^_pred_ was 0.66 (>0.6), and SEP of the test set was 0.16, which specified the model also had well predictive ability [42,43] and the constructed CoMSIA model met the requirements [44]. The model was used to predict the comprehensive evaluation value Z of PAE derivatives’ flammability, biotoxicity, and enrichment multi-effect. In addition, from the contribution rates of the molecular force fields in the CoMSIA model in Table 3, it could be seen that the contribution rates of S, E, H, D and A were 22.9%, 14.0%, 44.9%, 0.00%, and 11.2% separately, which indicated the spatial effect, electrical distribution, and hydrophobicity of the groups had a large impact on PAEs’ flammability, biotoxicity, and enrichment effects. In contrast, the effect of the hydrogen bond donor field was negligible.

In order for further verification, the flame retardancy of PAE derivatives, and a 3D-QSAR model of PAEs’ flammability single-effect was also established in this paper. In addition, single-effect 3D-QSAR models of PAEs’ biotoxicity, enrichment, degradability, and mobility were established to evaluate the eco-friendliness of PAE derivatives. In accordance with the above steps, the 3D-QSAR model of PAEs’ flammability single-effect was constructed and tested by selecting 13 PAEs arbitrarily as the training set, while the remaining four PAEs were considered as the test set; DMP used as the template molecule. Finally, the CoMSIA model that met the requirements was obtained. The optimal principal component *n* of the CoMSIA model was 10, the cross-validation coefficient q^2^ was 0.580 (>0.5), the non-cross-validation coefficient r^2^ was 1.000 (r^2^ > 0.9), SEE was 0.010, and F value was 9275.112, which indicated the model had good fitting ability and robustness. The external test coefficient r^2^_pred_ was 0.901 (>0.6), and SEP was 0.346, indicating that the model had a good predictive ability. So, the constructed CoMSIA model could be used to predict the flammability of PAEs derivatives.

In addition, PAEs’ single-effect CoMSIA models of biotoxicity, enrichment, degradability, and mobility were also established in the research sequence, which were used to test the eco-friendliness of the designed molecules. The 3D-QSAR model of PAEs’ biotoxicity single-effect was established by considering PAEs’ molecular structure parameters as independent variables and log*LC*_50_ of PAEs on fish as the dependent variables. The 3D-QSAR model of PAEs’ enrichment single-effect was established by considering PAEs’ molecular structures as independent variables and log*BCF* of PAEs as the dependent variables. The 3D-QSAR model of PAEs’ degradability single-effect was established, by considering PAEs’ molecular structure information as the independent variable and the degradable activity data (log*HL*) of PAEs as the dependent variable. The 3D-QSAR model of PAEs’ mobility single-effect was established by considering PAEs’ molecular structure parameters as the independent variables and log*K*_OA_ of PAEs as the dependent variables. By evaluating the above parameters, it may be granted that the constructed models met the requirements and had good predictive abilities (Table 4). The contribution rates of the models’ molecular force fields are shown in Table 5. The experimental values, predicted values, and relative errors of the molecules in the model training set and test set are given in Table 6.

### 3.2. Molecular Modification and Evaluation Based on PAEs’ 3D-QSAR Model

To obtain flame retardant PAE derivatives with low biotoxicity and low enrichment, the CoMSIA model contour maps were generated by a multi-effect 3D-QSAR model of PAEs’ flammability, biotoxicity, and enrichment; molecular modification schemes for single and double substitutions of target molecules were formulated, by selecting DMP and DAP as target molecules. The three-dimensional contour maps explain the interaction of small molecules, the relationship between the molecules, and the receptors visibilities, which provide a basis for modifying compounds; the colorful blocks of the contour maps are beneficial to enhance molecular activity by default. Taking DMP as an example, the three-dimensional contour maps of the CoMSIA model were analyzed (Figure 1). The yellow block of the steric field indicated that the introduction of smaller groups in this area can improve the activity of DMP derivatives. Therefore, the introduction of bulky groups here was conducive to reduce the comprehensive effect value of the molecule, since PAEs’ comprehensive evaluation value was an inverse index. In the electrostatic field, the introduction of positively charged groups in the blue region, and the introduction of negatively charged groups in the red region are beneficial to improve the activity of the molecule. In the hydrophobic field, the introduction of hydrophilic groups in the white area helps to improve the activity of the molecule. In order to reduce the comprehensive evaluation value of DMP’s flammability, biotoxicity, and enrichment effects, hydrophobic groups were introduced into the molecule. In the hydrogen bond receptor field, the purple-colored block indicates that increasing the number of hydrogen bond receptors is beneficial to improve the activity of DMP, while the red areas indicate the increased number of hydrogen bond donors reduces the activity of molecules.

During the modification of the target molecules, only the steric field and hydrophobic field with the highest contribution rates in the molecular field were considered. The three-dimensional contour maps showed that the substitution sites are surrounded by the steric field and the hydrophobic field, which provided a reference for molecular modification of PAEs. It could be seen in Figure 1A that the yellow regions were distributed near the H atom connected to the 2-C atom and the 2′-C atom in the DMP and the white region in Figure 1C covered the side chain on the benzene ring of DMP, which indicated the introduction of bulky groups or hydrophobic groups at these sites could be capable of reducing the comprehensive evaluation value of PAEs’ flammability, biotoxicity, and enrichment multi-effect. When the molecules were modified, it was noteworthy that the basic skeleton part of PAEs should be retained, as well as retain their performance as a plasticizer. Based on the contour map information of the steric field and the hydrophobic field, the substitution sites of the target molecules DMP and DAP are shown in Figure 2.

Eight kinds of hydrophobic and bulky groups, such as –CH_3_, –CH_2_CH_3_, –CH_2_C_6_H_5_, –NO_2_, –CH_2_NO_2_, –SH, –OCH_3_, –CH=CH_2_, were selected to perform single and double position modifications of target molecules. Further, a total of 18 DMP derivatives and 20 DAP derivatives were designed.

The comprehensive evaluation value Z of 38 PAE derivatives was predicted using the 3D-QSAR model of PAEs’ flammability, biotoxicity, and enrichment multi-effect. LOI, log*LC*_50_, log*BCF* of PAE derivatives were predicted by PAEs’ flammability, biotoxicity, and enrichment single-effect 3D-QSAR models, respectively. Finally, by comparing with the target molecules, it was found that only 22 PAE derivatives had a lower comprehensive evaluation value, improved flammability, decreased biotoxicity, and decreased enrichment. The specific prediction results were shown in Table 7.

It could be seen from Table 7 that the 22 kinds of PAE derivatives had an increasing rate of LOI value of 0.74–9.12%, expressing their flame retardant properties were partially improved. The log*LC*_50_ values of PAE derivatives were reduced by one to three orders of magnitude in comparison to the target molecules, with a reduced rate of 9.50–338.22%. As for enrichment effect, the reduced degrees of log*BCF* of PAE derivatives were 10.79–90.99%. The log*BCF* values were all less than 3.30, indicating that the molecules would not be accumulated in the organisms [45]. When the LOI value remains 22–27%, molecules represent good flame retardant properties, and they are not easy to burn [28]. In order to obtain PAE derivatives with an obvious modification on the decreased flammability effect, seven PAE derivatives expressing flammability improvement rate >5% were selected for the subsequent analysis, such as DAP-2-CH_2_NO_2_, DAP-1-NO_2_-2-CH_2_C_6_H_5_, DAP-1-NO_2_-2-CH_2_CH_3_, DAP-1-NO_2_-2-CH_2_NO_2_, DAP-1-NO_2_-2-NO_2_, DAP-1-NO_2_-2-OCH_3_, and DAP-2-CH=CH_2_-1-NO_2_.

### 3.3. Evaluation of Eco-Friendliness, Stability, and Insulation of PAE Derivatives

The eco-friendliness (i.e., degradability, long-distance migration) and other properties (i.e., stability and insulation) of the PAE derivatives screened in Section 3.2 were further evaluated. We calculated log*HL* (representing degradability), log*K*_OA_ (representing long-distance mobility), Freq. (representing stability) and energy gap (representing insulation) of PAEs and their derivatives, and the results are given in Table 8.

The predictive results of PAEs’ degradability 3D-QSAR model showed that log*HL* values of all PAE derivatives were decreased, which explains the enhancement in their degradability. The log*HL* value of DAP-2-CH=CH_2_-1-NO_2_ was 11.84% lower than that of DAP. According to the predicted results of PAEs’ mobility 3D-QSAR model, although the log*K*_OA_ values of PAEs derivatives were reduced (except DAP-2-CH_2_NO_2_), all the log*K*_OA_ values of derivatives situated in a range of 6.5–10 with the same molecular mobility level, while all of them were semi-volatile substances [46]. In addition, all the positive frequencies Freq. of PAE derivatives were greater than 0, indicating that the designed molecules could exist stably in the environment. The insulativity of PAEs derivatives was reduced to some extent but within an acceptable range. It was confirmed that the six PAEs derivatives, DAP-1-NO_2_-2-CH_2_C_6_H_5_, DAP-1-NO_2_-2-CH_2_CH_3_, DAP-1-NO_2_-2-CH_2_NO_2_, DAP-1-NO_2_-2-NO_2_, DAP-1-NO_2_-2-OCH_3_, and DAP-2-CH=CH_2_-1-NO_2_, were eco-friendly plasticizers with flame retardancy.

### 3.4. Mechanism Analysis of PAE Derivatives’ Improved Multi-Effect and Flammability, Biotoxicity, Enrichment Effects 

#### 3.4.1. Mechanism Analysis of PAE Derivatives’ Improved Multiple Effects Based on the Three-Dimension Contour Maps 

In this paper, DMP was taken as an example to compare three-dimension contour maps of PAEs’ multi-effect model with flammability, biotoxicity, and enrichment single-effect models, respectively, so as to qualitatively analyze the mechanism of PAE derivatives’ improved flammability, biotoxicity, and enrichment comprehensive effect.

As can be seen from Table 5, the steric field and hydrophobic field had the maximum contribution to the force field contribution rate of PAEs’ multi-effect CoMSIA model and PAEs’ flammability, biotoxicity, enrichment single-effect models. The comprehensive evaluation value Z and the activity data of the multi-effect CoMSIA model were as small as possible, while the modified PAE derivatives showed reduced comprehensive evaluation value Z by changing the structure of the molecules. The characteristic values LOI of flammability and log*LC*_50_ of biotoxicity, both were positive indicators, while the larger values indicated the better results. The smaller values of log*BCF* indicated lower enrichment of molecules in organisms and thus represented the smaller environmental effects.

In Figure 3, for the steric field, a three-dimension contour map in the multi-effect model was shown in yellow, indicating that the introduction of small groups in the molecule would increase the comprehensive evaluation value. Therefore, in order to reduce the comprehensive evaluation value, larger groups should be introduced into the overlapping part of DMP with three-dimension contour maps during modification of the target molecules; this helped to improve the flammability, biotoxicity and enrichment effects of the molecule simultaneously. The modified information given by the three-dimension contour maps of the PAEs’ flammability and biotoxicity single-effect models indicated the introduction of large groups to DMP was beneficial to the increase of LOI and log*LC*_50_. So, the DMP derivatives modified by virtue of the comprehensive effect model information could simultaneously meet the requirements of increased flammability and reduced biotoxicity. According to the steric field three-dimension contour map of PAEs’ enrichment single-effect model, DMP did not overlap with any color areas, but obviously, the yellow block was bigger. Thus, the introduction of the small group was more conducive to improve the molecular activity; in other words, a large group would reduce the log*BCF* of the molecules to decrease the enrichment. Therefore, it could be concluded that the modified information of the steric field three-dimension contour maps of the enrichment single-effect model was also consistent with the comprehensive effect.

According to the contour maps of the hydrophobic field in Figure 3, the yellow region covered both sides of the DMP molecule in the contour map of PAEs’ multi-effect model, which indicated the introduction of the hydrophobic groups was beneficial to reduce the comprehensive evaluation value of the molecule. The contour map of the flammability single-effect model indicated that the introduction of the hydrophobic group at the end of the side chain of the DMP molecule could improve the LOI value, which could enhance the flame retardancy of the molecule. The yellow block was distributed in the side chain of the DMP molecule in the contour map of the biotoxicity model, indicating the introduction of hydrophobic groups could increase the log*LC*_50_ of the molecule to decrease the biotoxicity. In the contour map of enrichment model, the volume of the white block was larger, specifying that the introduction of hydrophobic groups was more conducive to improve the molecular activity; this information indicated that the addition of hydrophobic groups during molecular modification could reduce the enrichment.

#### 3.4.2. Mechanism Analysis of PAE Derivatives’ Improved Multiple Effects Based on Modified Group Properties

In addition to explaining the molecular modification results from the perspective of three-dimension contour maps, this paper also stated the properties of modified groups by revealing the different improvements on the PAEs’ flame retardation, biotoxicity, and enrichment effects; the study was performed by introducing groups with different properties. The reliability of the PAEs’ multi-effect 3D-QSAR model was proved further.

According to the contour map information of PAEs’ flammability, biotoxicity, and enrichment multi-effect 3D-QSAR model, the modified groups were introduced into PAE derivatives having the characteristics of large volume and hydrophobicity. The hydrophobic value of the group was expressed as the log*P* of the corresponding PAEs derivatives. The larger value of log*P* indicated higher hydrophobicity. The volume and hydrophobicity values of the modified groups were weighted and coupled to evaluate the strength of the group effect. The weight of the volume and the hydrophobicity of the modified group represented the contribution rate of the steric field and hydrophobic field in PAEs’ flammability, biotoxicity, and enrichment multi-effect model, respectively. Further, the coupling values of modified groups’ properties were calculated (Table 9).

The property of each modified group was represented by the average values of the property of coupling values and introduced in Table 9. The average reduction amplitude values of the comprehensive molecular effects, the average increase amplitude values of the flammability effect, the average reduction amplitude values of the biotoxicity, and the average reduction amplitude values of the enrichment of molecules with the same substituted group, were calculated further (Table 10). As shown in Table 10, the comprehensive effect improvement degree values of these groups (–CH_2_NO_2_, –CH_2_C_6_H_5_, –NO_2_) were superior to the other introduced groups. The coupling values of each group and the average reduction amplitude values of the comprehensive effect were analyzed by linear regression; in the obtained results, the correlation coefficient R = 0.7516 > 0.7067 (P = 0.05, R = 0.7067), indicated that the modification effect of PAE derivatives was significantly correlated with the volume and hydrophobicity of introduced groups. In addition, the average improvement degree values in flammability, biotoxicity, and enrichment of PAE derivatives were weighted with the weights of 40%: 30%: 30%, and thus the weighted improvement degree values of the comprehensive effect were obtained. The average reduction amplitude values of PAE derivatives’ comprehensive effect and the weighted improvement degree values of comprehensive effect were analyzed by linear regression, while the correlation coefficient R of them was 0.8781; in the obtained result, there was a significant correlation at P = 0.01 (R = 0.8343). By virtue of the above results, it was clear that the comprehensive evaluation values of PAEs’ flammability, biotoxicity and enrichment effects calculated by the comprehensive evaluation method in this paper could represent the multiple effects of PAEs simultaneously, which confirmed the reliability of the 3D-QSAR model of PAEs’ flammability, biotoxicity, and enrichment multi-effect.

## 4. Conclusions

Using the ideal point method, an improved flammability, biotoxicity, and enrichment multi-effect 3D-QSAR model for PAEs was developed in this paper, and it was successfully applied to the molecular modification of plastic additives PAEs that combined flame retardancy with improvements in biotoxicity and enrichment. Finally, six eco-friendly molecules (DAP-2-CH_2_NO_2_, DAP-1-NO_2_-2-CH_2_C_6_H_5_, DAP-1-NO_2_-2-CH_2_CH_3_, DAP-1-NO_2_-2-CH_2_NO_2_, DAP-1-NO_2_-2-NO_2_, DAP-1-NO_2_-2-OCH_3_, and DAP-2-CH=CH_2_-1-NO_2_) were obtained, with an increasing rate of LOI value of 0.74–9.12%. Their LOI was between 22–27, which indicates that they were equipped with flame retardancy technically. Their biotoxicity for fish was decreased to a great extent, because the log*LC*_50_ of six molecules was reduced with a range from 127.81–338.22%. Meanwhile, the log*BCF* values were all less than 3.30 and the molecules would not be accumulated in the organisms. The six molecules’ degradability and migration were slightly improved or almost remained unchanged.

In this paper, the reasons for the simultaneous improvement of multiple effects of derivatives were analyzed. Based on the modified information of contour map of PAEs’ multi-effect and single-effect 3D-QSAR models for flammability, biotoxicity, and enrichment, it was confirmed that the introduction of large groups and hydrophobic groups may be beneficial to the simultaneous improvement of PAEs’ comprehensive effects, and multiple effects of flammability, biotoxicity, and enrichment. In addition, the coupling values of the properties of modified groups were calculated by the volumes of weighting groups and the hydrophobic values; the calculation was performed according to the rate of steric and hydrophobic force field contribution in PAEs’ flammability, biotoxicity, and enrichment multi-effect model. The results showed that the coupling values of group properties were directly proportional to the improved degrees of the comprehensive effects of corresponding PAE derivatives, and also correlated significantly (R = 0.7516 > 0.7067 (P = 0.05, R = 0.7067)). The results confirmed the feasibility of the multi-effect 3D-QSAR model of PAEs’ flammability, biotoxicity, and enrichment effects.

However, the flame retardancy of PAEs derivatives designed in this paper still had some room to improve, and further attention should be paid to the group types and key quantitative parameters that may affect the flammability performance of PAEs.

## Figures and Tables

**Figure 1 polymers-12-01942-f001:**
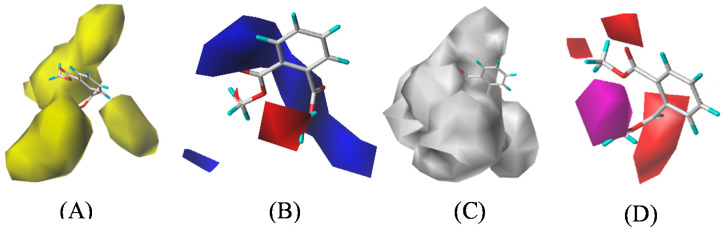
Dimethyl phthalate’s (DMP’s) three-dimensional contour maps of the steric field (**A**), electrostatic field (**B**), hydrophobic field (**C**), and hydrogen bond acceptor field (**D**) in the multi-effect CoMSIA model.

**Figure 2 polymers-12-01942-f002:**
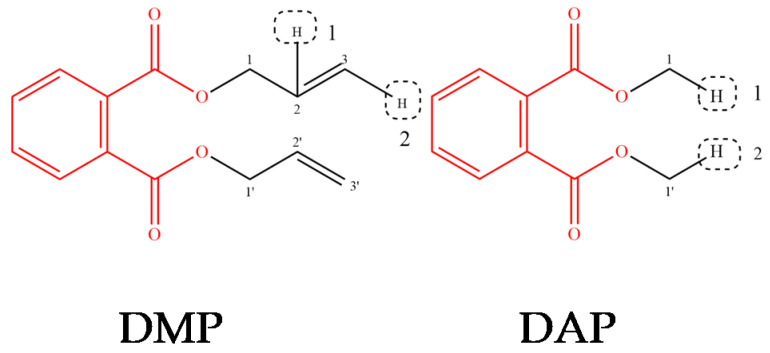
Modification sites of target molecules DMP and diallyl phthalate (DAP).

**Figure 3 polymers-12-01942-f003:**
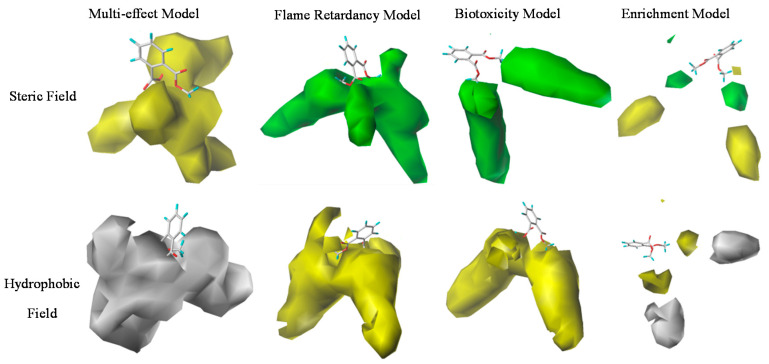
DMP’s three-dimensional contour maps for the multi-effect model and single-effect models of PAEs’ flammability, biotoxicity, and enrichment.

**Table 1 polymers-12-01942-t001:** Limited oxygen index (LOI), LC_50_, log*BCF*, log*HL*, log*K*_OA_ parameters of 17 phthalic acid esters (PAEs) and their comprehensive evaluation value Z for flammability, biotoxicity, and enrichment.

Molecule	LOI (%)	Fish LC_50_ (mg/L)	log*BCF*	log*HL*	log*K*_OA_	Z
dimethyl phthalate (DMP)	22.44	40.822	0.723	3.617	6.694	0.371
diisopropyl phthalate (DIPrP)	21.34	4.568	1.534	3.179	7.376	0.599
diethyl phthalate (DEP)	21.82	12.471	1.264	3.156	7.505	0.526
di-*n*-butyl phthalate (DBP)	20.95	1.113	2.636	2.734	8.631	0.712
di-isopentyl phthalate (DIPP)	20.63	0.398	3.260	2.904	9.504	0.794
benzyl butyl phthalate (BBP)	20.57	0.911	2.788	2.915	9.018	0.760
diisobutyl phthalate (DIBP)	20.95	1.356	2.379	3.182	8.412	0.692
di-*n*-octyl phthalate (DNOP)	19.96	0.008	2.988	2.655	12.079	0.845
diundecyl phthalate (DUP)	19.52	0.000183	1.330	1.398	14.068	0.818
di-isoheptyl phthalate (DIHP)	20.15	0.028	3.255	2.501	11.122	0.844
ditridecyl phthalate (DTDP)	19.31	0.0000146	0.964	0.924	15.535	0.841
di-*n*-hexyl phthalate (DHP)	20.37	0.095	2.793	1.639	9.799	0.784
di-*n*-pentyl phthalate (DPP)	20.63	0.327	2.001	3.063	9.674	0.703
diallyl phthalate (DAP)	21.40	5.323	1.798	3.895	8.032	0.605
di-iso-hexyl-phthalates (DIHxP)	20.37	0.116	3.908	2.623	10.239	0.880
di-*n*-propyl phthalate (DPrP)	21.34	3.749	1.825	3.362	8.053	0.617
dimethoxyethyl phthalate (DMEP)	20.90	124.130	0.402	9.544	9.766	0.311

**Table 2 polymers-12-01942-t002:** Comprehensive evaluation value Z of PAEs’ flammability, biotoxicity, and enrichment multi-effect, predicted values (Pred.) and relative errors.

Molecule	Z	Pred.	Relative Error (%)
DMP ^a^	0.371	0.369	−0.54
DIPrP ^a^	0.599	0.599	0.00
DEP ^b^	0.526	0.563	7.03
DBP ^a^	0.712	0.703	−1.26
DIPP ^b^	0.794	0.583	−26.57
BBP ^a^	0.760	0.760	0.00
DIBP ^a^	0.692	0.692	0.00
DNOP ^b^	0.845	0.752	−11.01
DUP ^a^	0.818	0.818	0.00
DIHP ^a^	0.844	0.844	0.00
DTDP ^b^	0.841	0.814	−3.21
DHP ^a^	0.784	0.784	0.00
DPP ^a^	0.703	0.707	0.57
DAP ^a^	0.605	0.605	0.00
DIHxP ^a^	0.880	0.880	0.00
DPrP ^a^	0.617	0.624	1.13
DMEP ^a^	0.311	0.311	0.00

^a^ Training set; ^b^ Test set.

**Table 3 polymers-12-01942-t003:** Evaluation parameters and molecular field contribution rate of PAEs’ flammability, biotoxicity, and enrichment multi-effect CoMSIA model.

Model	*n*	q^2^	r^2^	SEE	F	r^2^_pred_	SEP	S	E	H	D	A
CoMSIA	10	0.616	1	0.008	507.314	0.66	0.16	22.90%	14.00%	44.90%	0.00%	11.20%

**Table 4 polymers-12-01942-t004:** 3D-QSAR models’ parameters of PAEs’ flammability, biotoxicity, enrichment, degradability, and mobility single-effect.

3D-QSAR Model	Template Molecule	q^2^	*n*	r^2^	F	SEE	r^2^_pred_	SEP
Flammability	DMP	0.580	10	1.000	9275.112	0.010	0.901	0.346
Biotoxicity	DTDP	0.832	6	0.999	1375.319	0.077	0.997	0.169
Concentration	DIHxP	0.526	6	0.999	673.700	0.054	0.868	1.096
Degradability	DAP	0.747	4	0.958	39.680	0.222	0.724	0.494
Mobility	DTDP	0.756	4	0.998	162.553	0.342	0.993	0.311

**Table 5 polymers-12-01942-t005:** Molecular force field contribution rates of PAEs’ flammability, biotoxicity, enrichment, degradability, and mobility models.

3D-QSAR Model	S (%)	E (%)	H (%)	D (%)	A (%)
Flammability	30.7	15.9	39.6	0	13.7
Biotoxicity	31.0	13.2	44.4	0.0	11.3
Concentration	32.4	15.3	41.9	0.0	10.3
Degradability	35.7	12.9	47.5	0.0	3.9
Mobility	32.5	13.7	42.1	0.0	11.6

**Table 6 polymers-12-01942-t006:** The log*LC*_50_, log*BCF*, log*HL*, and log*K*_OA_ values of the training sets and test sets in PAEs’ single-effect models, and their Pred. and relative errors.

Flammability Model				Biotoxicity Model				Concentration Model				Degradability Model				Mobility Model			
Molecule	LOI (%)	Pred.	Relative Error (%)	Molecule	log*LC*_50_	Pred.	Relative Error (%)	Molecule	log*BCF*	Pred.	Relative Error (%)	Molecule	log*HL*	Pred.	Relative Error (%)	Molecule	log*K*_OA_	Pred.	Relative Error (%)
DMP ^a^	22.44	22.440	0.00	DMP ^a^	1.61	1.601	−0.56	DMP ^a^	0.723	0.742	2.63	DMP ^a^	3.617	3.622	0.14	DMP ^a^	6.694	6.737	0.64
DIPrP ^a^	21.34	21.341	0.00	DIPrP ^a^	0.66	0.679	2.88	DIPrP ^a^	1.534	1.573	2.54	DIPrP ^a^	3.179	3.381	6.35	DIPrP ^a^	7.376	7.66	3.85
DEP ^a^	21.82	21.820	0.00	DEP ^a^	1.10	1.075	−2.27	DEP ^a^	1.264	1.248	−1.27	DEP ^b^	3.156	3.363	6.56	DEP ^b^	7.505	7.72	2.86
DBP ^a^	20.95	20.943	−0.03	DBP ^a^	0.05	0.105	110.00	DBP ^b^	2.636	2.333	−11.49	DBP ^a^	2.734	3.073	12.40	DBP ^b^	8.631	8.548	−0.96
DIPP ^b^	20.63	20.778	0.72	DIPP ^a^	−0.40	−0.428	7.00	DIPP ^a^	3.26	3.259	−0.03	DIPP ^b^	2.904	3.251	11.95	DIPP ^a^	9.504	9.597	0.98
BBP ^a^	20.57	20.570	0.00	BBP ^a^	−0.04	−0.053	32.50	BBP ^a^	2.788	2.798	0.36	BBP ^b^	2.915	3.266	12.04	BBP ^b^	9.018	9.074	0.62
DIBP ^a^	20.95	20.950	0.00	DIBP ^a^	0.13	0.130	0.00	DIBP ^a^	2.379	2.403	1.01	DIBP ^a^	3.182	3.137	−1.41	DIBP ^a^	8.412	8.607	2.32
DNOP ^b^	19.96	20.124	0.82	DNOP ^a^	−2.10	−2.102	0.10	DNOP ^b^	2.988	2.532	−15.26	DNOP ^a^	2.655	2.787	4.97	DNOP ^a^	12.079	11.845	−1.94
DUP ^a^	19.52	19.512	−0.04	DUP ^a^	−3.74	−3.850	2.94	DUP ^a^	1.33	1.363	2.48	DUP ^a^	1.398	1.309	−6.37	DUP ^b^	14.068	14.352	2.02
DIHP ^a^	20.15	20.150	0.00	DIHP ^a^	−1.55	−1.556	0.39	DIHP ^a^	3.255	3.226	−0.89	DIHP ^a^	2.501	2.408	−3.72	DIHP ^a^	11.122	10.857	−2.38
DTDP ^a^	19.31	19.316	0.03	DTDP ^a^	−2.10	−2.102	0.10	DTDP ^a^	0.964	0.932	−3.32	DTDP ^a^	0.924	0.976	5.63	DTDP ^a^	15.535	15.427	−0.70
DHP ^a^	20.37	20.376	0.03	DHP ^a^	−1.02	−1.066	4.51	DHP ^a^	2.793	2.84	1.68	DHP ^b^	1.639	2.543	55.16	DHP ^a^	9.799	10.219	4.29
DPP ^b^	20.63	20.622	−0.04	DPP ^b^	−0.49	−0.469	−4.29	DAP ^b^	1.798	1.835	2.06	DPP ^a^	3.063	2.82	−7.93	DPP ^a^	9.674	9.377	−3.07
DAP ^a^	21.40	21.399	0.00	DAP ^b^	0.73	0.504	−30.96	DIHxP ^a^	3.908	3.894	−0.36	DAP ^a^	3.895	3.608	−7.37	DAP ^b^	8.032	8.079	0.59
DIHxP ^b^	20.37	20.784	2.03	DPrP ^b^	0.57	0.671	17.72	DPrP ^a^	1.825	1.745	−4.38	DIHxP ^a^	2.623	2.7	2.94	DIHxP ^a^	10.239	9.886	−3.45
DPrP ^a^	21.34	21.344	0.02	DMEP ^a^	2.09	2.134	2.11					DPrP ^a^	3.362	3.31	−1.55	DPrP ^a^	8.053	7.707	−4.30
DMEP ^a^	20.90	20.900	0.00													DMEP ^a^	9.766	10.049	2.90

^a^ Training set; ^b^ Test set.

**Table 7 polymers-12-01942-t007:** Comprehensive evaluation value Z, LOI, log*LC*_50_, log*BCF* of PAEs derivatives based on 3D-QSAR models and their degree of change.

Molecule	Z	Decreasing (%)	LOI	Increasing (%)	log*LC*_50_	Increasing (%)	log*BCF*	Decreasing (%)
DMP	0.371	-	22.44	-	1.61	-	0.723	-
DMP-1-CH_2_NO_2_	0.085	77.09	23.39	4.23	1.76	9.50	0.161	77.73
DMP-1-NO_2_	0.234	36.93	22.92	2.13	1.98	22.67	0.372	48.55
DMP-1-CH_3_-2-CH_2_NO_2_	0.145	60.92	23.12	3.03	2.29	42.48	0.353	51.18
DMP-1-CH_3_-2-NO_2_	0.292	21.29	22.66	0.98	2.17	34.97	0.556	23.10
DAP	0.605	-	21.40	-	0.73	-	1.798	-
DAP-1-CH_2_NO_2_	0.325	46.28	22.23	3.86	2.28	212.47	1.007	43.99
DAP-1-SH	0.519	14.21	21.82	1.94	1.13	54.52	1.360	24.36
DAP-2-CH_2_NO_2_	0.279	53.88	22.52	5.24	2.27	210.41	0.162	90.99
DAP-2-NO_2_	0.293	51.57	22.36	4.50	1.94	165.89	0.183	89.82
DAP-2-SH	0.557	7.93	21.64	1.13	0.94	28.63	1.604	10.79
DAP-1-NO_2_-2-CH_2_C_6_H_5_	0.301	50.25	22.65	5.83	1.66	127.81	1.182	34.26
DAP-1-NO_2_-2-CH_2_CH_3_	0.365	39.67	22.57	5.49	1.88	157.95	1.482	17.58
DAP-1-NO_2_-2-CH_2_NO_2_	0.164	72.89	23.18	8.30	2.55	249.45	1.324	26.36
DAP-1-NO_2_-2-CH_3_	0.471	22.15	21.65	1.15	0.92	26.30	0.830	53.84
DAP-1-NO_2_-2-CH=CH_2_	0.447	26.12	22.18	3.63	2.03	177.81	0.696	61.29
DAP-1-NO_2_-2-NO_2_	−0.172	128.43	23.35	9.12	3.20	338.22	0.761	57.68
DAP-1-NO_2_-2-OCH_3_	0.340	43.80	22.54	5.32	1.83	150.96	1.401	22.08
DAP-1-NO_2_-2-SH	0.497	17.85	21.72	1.47	0.96	30.96	1.557	13.40
DAP-2-CH=CH_2_-1-CH_2_NO_2_	0.353	41.65	21.63	1.09	0.95	30.41	0.898	50.06
DAP-2-CH=CH_2_-1-CH_3_	0.529	12.56	21.56	0.74	0.90	23.70	1.366	24.03
DAP-2-CH=CH_2_-1-NO_2_	0.399	34.05	22.53	5.27	1.67	128.49	1.166	35.15
DAP-2-CH=CH_2_-1-OCH_3_	0.602	0.50	21.79	1.80	0.94	28.08	1.393	22.53
DAP-2-CH=CH_2_-1-SH	0.543	10.25	21.59	0.89	0.84	14.38	1.459	18.85

**Table 8 polymers-12-01942-t008:** Predicted log*HL* and log*K*_OA_ values based on PAEs’ degradability, mobility 3D-QSAR models, Freq., and energy gap values based on Gaussian calculation.

Molecule	log*HL*	Decreasing (%)	log*K*_OA_	Increasing (%)	Freq.	Energy Gap (a.u.)	Increasing (%)
DAP	3.895	0.00	8.032	0.00	17.30	0.203	0.00
DAP-2-CH_2_NO_2_	3.875	0.51	6.282	−21.79	9.38	0.195	−3.89
DAP-1-NO_2_-2-CH_2_C_6_H_5_	3.643	6.47	6.902	−14.07	12.33	0.154	−23.84
DAP-1-NO_2_-2-CH_2_CH_3_	3.678	5.57	7.121	−11.34	12.94	0.156	−22.97
DAP-1-NO_2_-2-CH_2_NO_2_	3.624	6.96	6.936	−13.65	7.75	0.154	−23.91
DAP-1-NO_2_-2-NO_2_	3.620	7.06	7.087	−11.77	21.69	0.127	−37.42
DAP-1-NO_2_-2-OCH_3_	3.527	9.45	7.523	−6.34	11.70	0.175	−13.40
DAP-2-CH=CH_2_-1-NO_2_	3.434	11.84	7.748	−3.54	20.13	0.162	−20.26

**Table 9 polymers-12-01942-t009:** Coupling values of the modified groups’ properties in PAE derivatives.

		Site 1					Site 2					Site 1 and 2				
Group	Molecule	Volume	Increasing (%)	log*P*	Increasing (%)	Coupling Value (%)	Volume	Increasing (%)	log*P*	Increasing (%)	Coupling Value (%)	Volume	Increasing (%)	log*P*	Increasing (%)	Coupling Value (%)
	DMP	1.0	0.00	1.54	0	0.00	1.0	0.00	1.54	0.00	0.00	2.0	0.00	1.54	0.00	0.00
	DAP	1.0	0.00	3.06	0	0.00	1.0	0.00	3.06	0.00	0.00	2.0	0.00	3.06	0.00	0.00
–CH_3_	DMP-1-CH_3_-2-CH_2_NO_2_	15.0	1400.00				60.0	5900.00				75.0	3650.00	1.30	−15.58	828.85
	DMP-1-CH_3_-2-NO_2_	15.0	1400.00				46.0	4500.00				61.0	2950.00	0.84	−45.45	655.14
	DAP-1-NO_2_-2-CH_3_	46.0	4500.00				15.0	1400.00				61.0	2950.00	2.77	−9.48	671.29
	DAP-2-CH=CH_2_-1-CH_3_	27.0	2600.00				15.0	1400.00				42.0	2000.00	3.75	22.55	468.12
–CH_2_CH_3_	DAP-1-NO_2_-2-CH_2_CH_3_	46.0	4500.00				29.1	2810.00				75.1	3655.00	3.18	3.92	838.76
–CH_2_C_6_H_5_	DAP-1-NO_2_-2-CH_2_C_6_H_5_	46.0	4500.00				91.1	9010.00				137.1	6755.00	4.37	42.81	1566.12
–NO_2_	DMP-1-NO_2_	46.0	4500.00	0.56	−63.64	1001.93										
	DMP-1-CH_3_-2-NO_2_	15.0	1400.00				46.0	4500.00				61.0	2950.00	0.84	−45.45	655.14
	DAP-2-NO_2_						46.0	4500.00	2.93	−4.25	1028.59					
	DAP-1-NO_2_-2-CH_2_C_6_H_5_	46.0	4500.00				91.1	9010.00				137.1	6755.00	4.37	42.81	1566.12
	DAP-1-NO_2_-2-CH_2_CH_2_CH_3_	46.0	4500.00				43.1	4210.00				89.1	4355.00	3.60	17.65	1005.22
	DAP-1-NO2-_2_-CH2CH_3_	46.0	4500.00				29.1	2810.00				75.1	3655.00	3.18	3.92	838.76
	DAP-1-NO_2_-2-CH_2_NO_2_	46.0	4500.00				60.0	5900.00				106.0	5200.00	2.38	−22.22	1180.82
	DAP-1-NO_2_-2-CH_3_	46.0	4500.00				15.0	1400.00				61.0	2950.00	2.77	−9.48	671.29
	DAP-1-NO_2_-2-CH=CH_2_	46.0	4500.00				27.0	2600.00				73.0	3550.00	2.91	−4.90	810.75
	DAP-1-NO_2_-2-NO_2_	46.0	4500.00				46.0	4500.00				92.0	4500.00	2.28	−25.49	1019.05
	DAP-1-NO_2_-2-OCH_3_	46.0	4500.00				31.0	3000.00				77.0	3750.00	1.96	−35.95	842.61
	DAP-1-NO_2_-2-SH	46.0	4500.00				33.1	3210.00				79.1	3855.00	2.43	−20.59	873.55
	DAP-2-CH=CH_2_-1-NO_2_	27.0	2600.00				46.0	4500.00				73.0	3550.00	2.91	−4.90	810.75
–CH_2_NO_2_	DMP-1-CH_2_NO_2_	60.0	5900.00	1.01	−34.42	1335.65										
	DMP-1-CH_3_-2-CH_2_NO_2_	15.0	1400.00				60.0	5900.00				75.0	3650.00	1.30	−15.58	828.85
	DAP-1-CH_2_NO_2_	60.0	5900.00	2.85	−6.86	1348.02										
	DAP-2-CH_2_NO_2_						60.0	5900.00	3.04	−0.65	1350.81					
	DAP-1-NO_2_-2-CH_2_NO_2_	46.0	4500.00				60.0	5900.00				106.0	5200.00	2.38	−22.22	1180.82
	DAP-2-CH=CH_2_-1-CH_2_NO_2_	27.0	2600.00				60.0	5900.00				87.0	4250.00	3.37	10.13	977.80
–SH	DAP-1-SH	33.1	3210.00	2.55	−16.67	727.61										
	DAP-2-SH						33.1	3210.00	3.09	0.98	735.53					
	DAP-1-NO_2_-2-SH	46.0	4500.00				33.1	3210.00				79.1	3855.00	2.43	−20.59	873.55
	DAP-2-CH=CH_2_-1-SH	27.0	2600.00				33.1	3210.00				60.1	2905.00	3.07	0.33	665.39
–OCH_3_	DAP-1-NO_2_-2-OCH_3_	46.0	4500.00				31.0	3000.00				77.0	3750.00	1.96	−35.95	842.61
	DAP-2-CH=CH_2_-1-OCH_3_	27.0	2600.00				31.0	3000.00				58.0	2800.00	2.60	−15.03	634.45
–CH=CH_2_	DAP-1-NO_2_-2-CH=CH_2_	46.0	4500.00				27.0	2600.00				73.0	3550.00	2.91	−4.90	810.75
	DAP-2-CH=CH_2_-1-CH_2_NO_2_	27.0	2600.00				60.0	5900.00				87.0	4250.00	3.37	10.13	977.80
	DAP-2-CH=CH_2_-1-CH_3_	27.0	2600.00				15.0	1400.00				42.0	2000.00	3.75	22.55	468.12
	DAP-2-CH=CH_2_-1-NO_2_	27.0	2600.00				46.0	4500.00				73.0	3550.00	2.91	−4.90	810.75
	DAP-2-CH=CH_2_-1-OCH_3_	27.0	2600.00				31.0	3000.00				58.0	2800.00	2.60	−15.03	634.45
	DAP-2-CH=CH_2_-1-SH	27.0	2600.00				33.1	3210.00				60.1	2905.00	3.07	0.33	665.39

**Table 10 polymers-12-01942-t010:** Average improvement degree of comprehensive effect and multiple single-effect of 8 modified groups.

Group	Comprehensive Effect (%)	Group Properties (%)	Flame Retardancy (%)	Biotoxicity (%)	Concentration (%)	Weighted Comprehensive Effect (%)
–CH_3_	29.23	655.85	1.48	31.86	38.04	21.56
–CH_2_CH_3_	39.67	838.76	5.49	157.95	17.58	54.86
–CH_2_C_6_H_5_	50.25	1566.12	5.83	127.81	34.26	50.95
–NO_2_	45.42	941.61	4.43	134.29	40.26	54.14
–CH_2_NO_2_	58.79	1170.32	4.29	125.79	56.72	56.47
–SH	12.56	750.52	1.36	32.12	16.85	15.23
–OCH_3_	22.15	738.53	3.56	89.52	22.31	34.97
–CH=CH_2_	20.86	727.88	2.24	67.15	35.32	31.63

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
