# Peer review of "A Modified 3D-QSAR Model Based on Ideal Point Method and Its Application in the Molecular Modification of Plasticizers with Flame Retardancy and Eco-Friendliness"

_polymers, 2020, doi:10.3390/polym12091942_

Round 1

Reviewer 1 Report

"A modified 3D-QSAR model based on ideal point method and its application in the molecular modification of plastic additive PAEs’ flame retardancy with eco-friendliness

Haigang Zhang *, Chengji Zhao * and Hui Na *

Alan G. MacDiarmid Institute, College of Chemistry, Jilin University, No. 2699 Qianjin Street, Changchun 7 City, Jilin Province, PR China

A 3D-QSAR model was developed for a set of 17 molecules of phthalic acid esters (PAE) to evaluate retardancy, biotoxicity, and enrichment properties. In addition, dimethyl phthalates (DMP) and diallyl phthalates (DAP) derivatives were tested as proposal to look for compounds with the best performance as plastic additives.

The manuscript is well written and descriptive with a methodology that is well established but solid. The expression “flame retardancy” is referred to in the text 57 times driving the readers to think that the PAEs and their derivatives are flame retardant compounds which is wrong. Flame retardants are well stablished diverse group of chemicals (organohalogens, organophosphorus, etc.) added to the plastic and other materials in the process of manufacture to prevent he fire or to slow the spread of the fire. Thus, PAEs are plasticizers but technically flame retardant. Therefore, the suggestion, it is to use another word to express the same idea because enlist PAEs as a flame retardancy is confused.

Manuscript needs to pay attention some quality problems and revision that must be addressed before it is accepted for publication in polymer.

Minor Specific comments

-    Line 132. It should be included on reference list entry or the website for the source “EPIWEB database”

-    Lines 142-143. Check the text in these lines that is splitted.

-    Line 163. Equation 5 is a general expression. It should be replaced by the Euclidian distance equation which is the expression for P =2.

-    Line 539-540. The reference should be replaced for another in English because it is in Chinese. There is a lot of references and great in English about Euclidian Distance"

Reviewer 2 Report

The work by Zhang and coworkers is a good candidate for publication in Polymers, but several points should be cared before I suggest publication of this manuscript:

  • The necessity behind doing this research is missing in the ABSTRACT. Which gap in the literature necessitated doing this work?
  • The usage of other methods for calculating the multi-effect based on the literature should be discussed in the INTRODUCTION. See the following research: "Flame retardancy index for thermoplastic composites." Polymers 11.3 (2019): 407. Description of complementary actions of mineral and organic additives in thermoplastic polymer composites by Flame Retardancy Index. Polymers for Advanced Technologies30(8), 2056-2066.
  • The symbol in the line of 142 and 143 should be modified.
  • In Materials and Methods, the source of 17 kinds of PAEs is not clear. The authors should explain the difference between different kinds of used PAEs by details.
  • The mentioned unit in line 174 should be corrected.
  • Conclusion is also very short and qualitative, not quantitative. Reader need to see some statistics to judge about results at a glimpse.

Based on above, my suggestion is “Major Revision”.

Reviewer 3 Report

Dear Authors,

congratulation to your paper.

I had only some formatting aspect

1) Figure 2 is on the next page

2) Reference 1 is between 23 and 24

BR

The reviewer

Round 2

Reviewer 2 Report

all comments have been considered. I suggest accept